# Optoelectronic Properties of Ti-doped SnO$_2$ Thin Films Processed under Different Annealing Temperatures

**Chi-Fan Liu [1], Chun-Hsien Kuo [2], Tao-Hsing Chen [1,\*] and Yu-Sheng Huang [1]**

[1] Department of Mechanical Engineering, National Kaohsiung University of Science and Technology, Kaohsiung 80778, Taiwan; i108142101@nkust.edu.tw (C.-F.L.); isu10207025a@gmail.com (Y.-S.H.)

[2] Department of Mold and Die Engineering, National Kaohsiung University of Science and Technology, Kaohsiung 80778, Taiwan; chkuo@nkust.edu.tw

\* Correspondence: thchen@nkust.edu.tw; Tel.: +886-7-3814526-15330

**Abstract:** Ti-doped SnO$_2$ transparent conductive oxide (TCO) thin films are deposited on glass substrates using a radio frequency (RF) magnetron sputtering system and then are annealed at temperatures in the range of 200–500 °C for 30 min. The effects of the annealing temperature on the structural properties, surface roughness, electrical properties, and optical transmittance of the thin films are then systematically explored. The results show that a higher annealing temperature results in lower surface roughness and larger crystal size. Moreover, an annealing temperature of 300 °C leads to the minimum electrical resistivity of $5.65 \times 10^{-3}$ Ω·cm. The mean optical transmittance increases with an increase in temperature and achieves a maximum value of 74.2% at an annealing temperature of 500 °C. Overall, the highest figure of merit ($\Phi_{TC}$) ($3.99 \times 10^{-4}$ Ω$^{-1}$) is obtained at an annealing temperature of 500 °C.

**Keywords:** SnO$_2$; Ti-doped; annealing temperature; electrical resistivity; transmittance

## 1. Introduction

Transparent conductive oxide (TCO) thin films possess excellent conductivity and optical transmittance in the visible and near-infrared regions, and are thus applied in many photoelectric components nowadays, including solar cells [1,2], organic light-emitting diodes [3,4], thin-film transistors [5,6], photovoltaic batteries [7–9], electrochromic devices [10–12], and tablet displays [13–16]. Metallic films are generally opaque in the visible light range. However, for film thicknesses of less than 100 Å, visible light is transmitted through the film, while infrared (IR) light is reflected. Moreover, for metals such as In$_2$O$_3$, ZnO, SnO$_2$, TiO$_2$, and CdO with energy gaps of 3 eV or more, the film also has excellent semiconducting properties [17].

The literature contains many studies on the optoelectronic properties of metallic films [18–20]. In addition, various authors have investigated the properties of three-layer TCO films with oxide/metal/oxide or metallic oxide/metal/metallic oxide structures [21]. The results have shown that such films not only suppress the reflection from the metallic layer in the visible light range but also produce a transmittance effect [22,23]. Consequently, the TCO thin films are used in solar cells, gas sensors, LCD displays, etc.

Among the various metal oxides in common use nowadays, SnO$_2$ has poorer electrical properties than ITO, but a superior photoelectric performance in the IR region. Furthermore, SnO$_2$ has good chemical and thermal stability and is also amenable to surface modification in order to expand its working wavelength range. As a result, SnO$_2$ conductive films are widely used for such applications

as gas sensors, solar energy battery electrodes, low-radiation glasses, etc. [24,25]. However, $SnO_2$ films are less easily used in tablet display applications due to their high electrical resistance and poor etching effect.

Accordingly, the present study explores the feasibility of improving the optical and electrical properties of $SnO_2$ thin films by doping the films with Ti. Note that Ti is deliberately selected as the dopant material here, since it has a maximum chemical valence of +4 [26], where the radius of $Ti^{4+}$ is 0.0605 nm, while that of $Sn^{4+}$ is 0.069 nm. Due to the similarity of the ion radii, the $Ti^{4+}$ ions readily replace the $Sn^{4+}$ ions in the crystal lattice of the $SnO_2$ and hence modify its electrical and optical behavior. The $Ti:SnO_2$ films are deposited on glass substrates using a radio frequency (RF) magnetron sputtering system and then are annealed at various temperatures in the range of 200–500 °C to prompt the diffusion of the Ti atoms into the $SnO_2$ layer. The optoelectronic properties of the films are then systematically explored in order to determine the annealing temperature, which results in the optimal tradeoff between the electrical and optical properties of the film, respectively.

## 2. Experimental Procedure

The glass plate was purchased from Corning company (Corning, NY, USA) and cut into pieces the size of 25 mm × 25 mm × 7 mm (length × width × thickness) using a diamond saw. The substrates were cleaned sequentially in deionized (DI) water, acetone, and IPA (isopropanol), and DI water once again in order to remove any pollutants, residual solvents, or nonorganic components from the substrate surface. The substrates were then dried in an oven at 90 °C until the water was completely vaporized. The thin films were prepared using a sputtering target (two-inch diameter) composed of $SnO_2$(95%) and $TiO_2$(5%). The $Ti:SnO_2$ films were then deposited on the glass substrates using an RF sputtering system with a sputtering power of 60 W, an argon gas flow rate of 29 sccm, an oxygen flow rate of 1 sccm, and a bias of 7.5 mTorr. The sputtering process was performed without substrate heating. The purpose of this experiment was to study the impact of different process parameters and conditions on the characteristics of a transparent conductive film. Following the deposition process, the $Ti:SnO_2$ films were annealed at temperatures of 200, 300, 400, and 500 °C for 30 min. The structures of the annealed thin films were examined by X-ray diffraction (XRD, Bruker, Billerica, MA, USA). In addition, the surface roughness and crystal size were determined by atomic force microscopy (AFM, NTMDT-AFM, Bruker, Billerica, MA, USA) and scanning electron microscopy (SEM, JEOL JSM-7000F, JEOL, Kyoto, Janpan), respectively. The photoelectric property data consisted of light transmittance, resistivity, carrier concentration by spectrophotometer (UV Spectrophotometer, Hitachi 2900, Hitachi, Tokyo, Japan), and a Hall measuring instrument(AHM-800B, Advnaced Design Technology, Taipei, Taiwan), respectively. In this paper, the measurement for each condition was performed six times to confirm the data.

## 3. Results and Discussion

### 3.1. Effects of Annealing Temperature on Ti:SnO₂ Film Thickness

Table 1 shows the thickness of the various $Ti:SnO_2$ films, measured by alpha–step profilometer (KLA-Tencor, Milpitas, CA, USA). In general, it is noted that while the annealing temperature has no significant effect on the film thickness, the thickness increases slightly in the film annealed at 200 °C but then reduces progressively as the annealing temperature is further increased to 500 °C.

**Table 1.** Effects of annealing temperature on thickness of Ti:SnO$_2$ films.

| Annealing Temperature (°C) | Thickness (nm) |
|---|---|
| as-deposited | 88.2 ± 2.0 |
| 200 | 93 ± 2.0 |
| 300 | 90 ± 2.0 |
| 400 | 89.4 ± 2.0 |
| 500 | 88.6 ± 2.0 |

*3.2. Effects of Annealing Temperature on Structural Properties of Ti:SnO$_2$ Films*

Figure 1 shows the XRD patterns of the as-deposited and annealed Ti:SnO$_2$ thin films. As the annealing temperature increases, prominent peaks are observed at 26.6°, 33.8°, and 51.7° corresponding to (110), (101), and (211) phases, respectively. The (101) phase is a combined SnO and SnO$_2$ phase [27]; some of the Sn and Ti atoms are replaced by a diffused process following the annealing process [28,29]. Notably, diffraction peaks are very small in the as-deposited film or the film annealed at 200 °C. However, as the annealing temperature increases to 300 °C, the crystalline phase appears more within the film. The crystalline structure becomes increasingly pronounced as the annealing temperature increases to 500 °C, and hence has a significant effect on the electrical and optical properties, as described in the following sections.

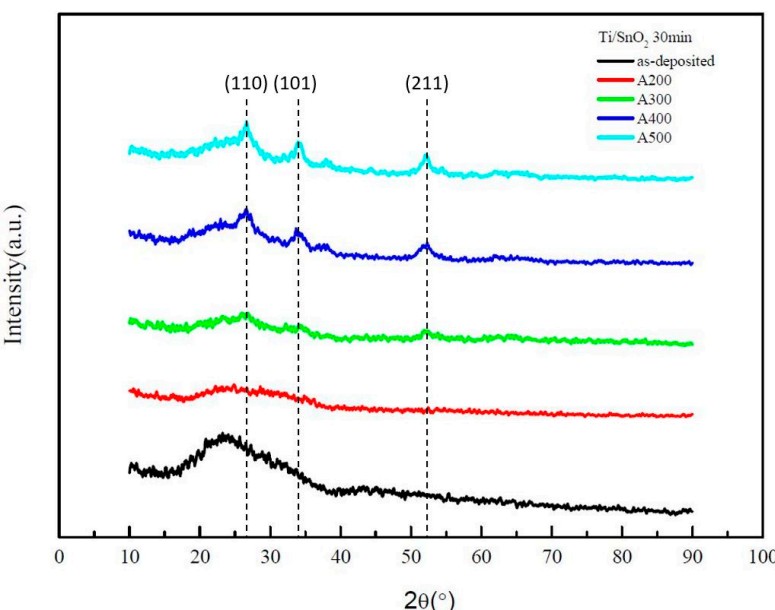

**Figure 1.** X-ray diffraction (XRD) patterns of as-deposited and annealed Ti:SnO$_2$ films.

*3.3. Effects of Annealing Temperature on Electrical Resistivity*

Figure 2 shows the electrical properties of the as-deposited and annealed Ti:SnO$_2$ films. Previous studies have shown that the TCO transmission mechanism is governed mainly by element doping and oxygen vacancies [29,30]. For the Ti:SnO$_2$ thin films considered in the present study, the oxygen vacancy contributes two free electrons, and therefore dominates the transmission mechanism. Although the Sn$^+$ atom also provides a free electron, it cannot be activated as effectively as the oxygen vacancy because the carrier concentration is primarily controlled by the oxygen vacancy. As described above, the Ti:SnO$_2$ film has a small crystal structure in the as-deposited condition and under an annealing temperature of 200 °C. However, as the annealing temperature is increased to 300 °C, the film has a low resistivity of $5.65 \times 10^{-3}$ Ω·cm as a result of the high carrier concentration. Meantime, as the

annealing temperature is increased beyond 300 °C, the SnO and SnO$_2$ combined phase are gradually formed, causing the carrier concentration to decrease and the resistivity to increase.

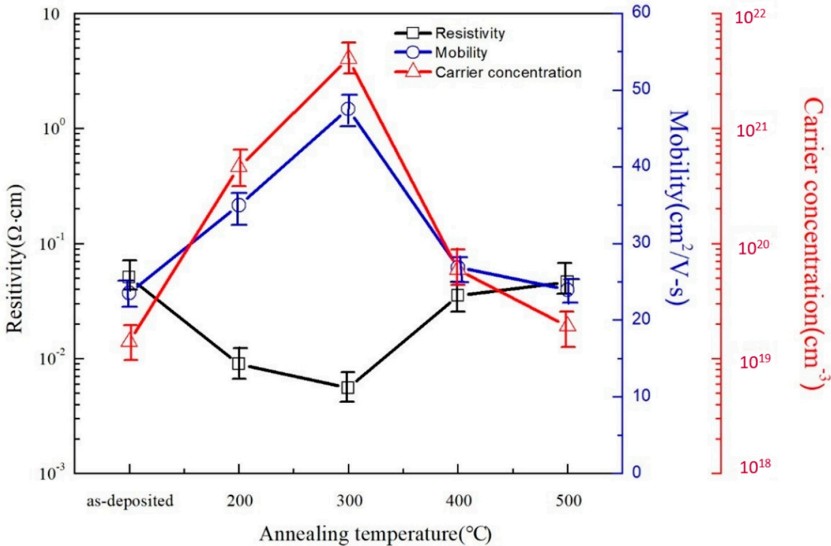

**Figure 2.** Electrical properties of as-deposited and annealed Ti:SnO$_2$ films.

*3.4. Transmittance*

Figure 3 shows the optical transmittance properties of the various Ti:SnO$_2$ thin films. For the as-deposited film, the mean transmittance has a low value of approximately 58% due to the poor effect of the replaced Sn for the Ti atom. However, the transmittance improves significantly in the annealed samples, particularly in those annealed at temperatures of 300 °C or more. In conventional ITO films, the optical energy gap theoretically increases as the carrier concentration increases, since the Fermi level moves into the conduction band and the electrons on the valence band are forced to jump to the conduction band, thereby requiring more energy and resulting in the so-called Burstein–Moss effect [31,32]. However, the optical energy gap rises with a decreasing carrier concentration. Such a phenomenon may be due to an interaction effect between ion compounds. For example, Zn$^{2+}$ and Sn$^{4+}$ ions coexist in IZTO films and trigger the generation of a donor-acceptor pair, which reduces the energy gap and mitigates the Burstein–Moss effect. For the present Ti:SnO$_2$ films, the carrier concentration decreases following annealing at temperatures higher than 200 °C (see Figure 2). However, the energy gap and mean transmittance both increase (see Table 2 and Figure 3, respectively). For an annealing temperature of 200 °C, the improvement in the transmittance is very modest (i.e., from around 58% for the as-deposited sample to approximately 60% for the annealed sample). However, for an annealing temperature of 300 °C, the film undergoes a transformation from a homogenous crystalline structure and the mean transmittance improves to almost 75%. Furthermore, as the annealing temperature increases, the transformation toward a crystalline structure becomes more complete (see Figure 1) and hence the mean transmittance increases. Thus, the film annealed at a temperature of 500 °C shows the maximum mean transmittance of approximately 74.2%.

**Table 2.** Effects of annealing temperature on energy gap (*Eg*) of Ti:SnO$_2$ films.

| Annealing Temperature (°C) | *Eg* (eV) |
|---|---|
| as-deposited | 2.95 |
| 200 | 2.88 |
| 300 | 3.13 |
| 400 | 3.21 |
| 500 | 3.28 |

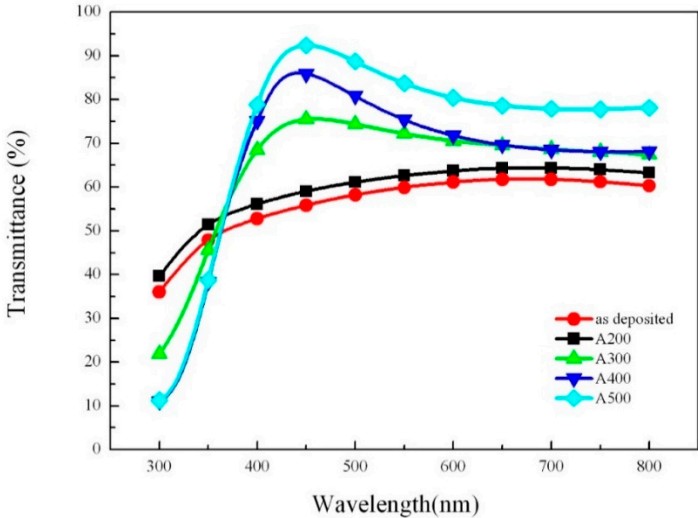

**Figure 3.** Optical transmittance of as-deposited and annealed Ti:SnO$_2$ films.

Figure 4 shows the relationship between the optical absorption coefficient ($\alpha$) and photon energy($h\nu$) for the Ti:SnO$_2$ film. The optical band gap (*Eg*) is calculated as follows with the equation [33,34]:

$$\alpha h\nu = A(h\nu - E_g)^{1/2} \tag{1}$$

where $\alpha$ is the absorption coefficient, $\nu$ is the frequency of incident light, h is the Planck's constant, and *A* is constant. The optical band gap is extrapolating the straight-line portion of the plot to the energy axis. Table 2 shows the calculated values of the optical band gap for the present Ti:SnO$_2$ thin films. Furthermore, an Eg value greater than 3 eV is regarded as excellent. Referring to Table 2, the Eg value of the present Ti:SnO$_2$ films increases with an increase in annealing temperature and is equal to 3.28 eV at an annealing temperature of 500 °C. Moreover, an Eg value greater than 3 eV is obtained for all of the films annealed at a temperature of 300 °C or more.

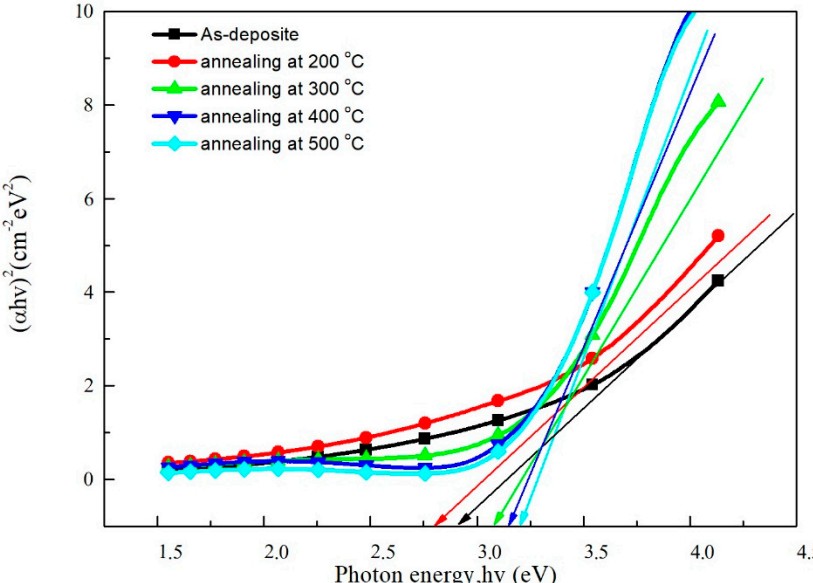

**Figure 4.** The ($\alpha h\nu)^2$ against photon energy ($h\nu$) of Ti:SnO$_2$ films under different annealing temperatures.

### 3.5. Surface Feature Analysis

Figures 5 and 6 show AFM and SEM images of the various as-deposited and annealed Ti:SnO$_2$ films. The mean surface roughness values of the films are listed in Table 3. As shown, the as-deposited sample has a surface roughness of 0.31 nm. However, following annealing at 300 °C, the surface roughness falls to a value of around 0.35 nm due to the formation of the crystalline phase. However, as the annealing temperature is further increased, the surface roughness reduces and has a value of just 0.296 nm in the sample processed at the highest annealing temperature of 500 °C. The AFM and SEM images show that the as-deposited Ti:SnO$_2$ film and the film annealed at 300 °C have higher surface roughness. Annealing at a temperature of 300 °C results in high roughness, but after 300 °C the roughness is decreased.

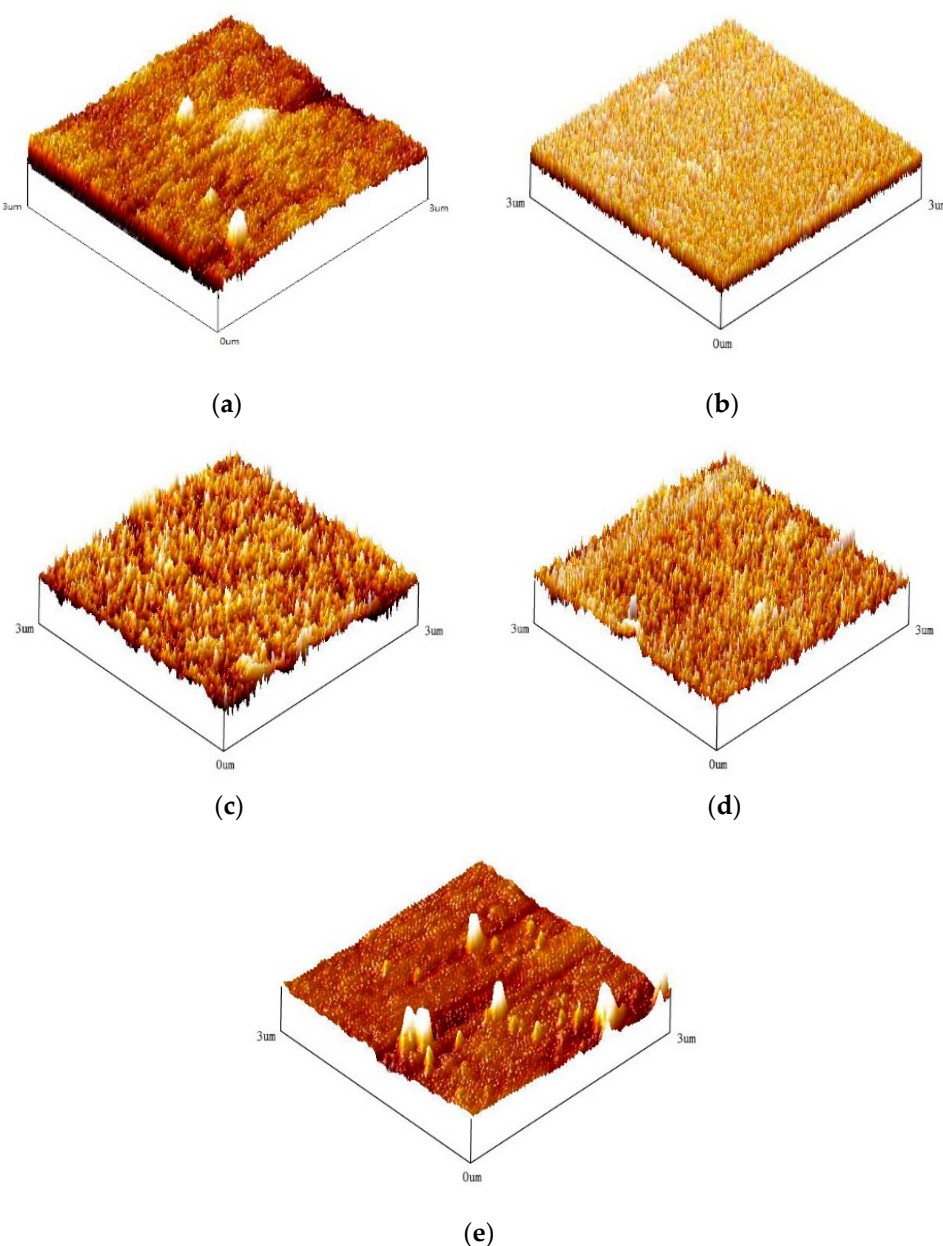

**Figure 5.** Atomic force microscopy (AFM) images of as-deposited and annealed Ti:SnO$_2$ films: (**a**) as-deposited, (**b**) annealed at 200 °C, (**c**) annealed at 300 °C, (**d**) annealed at 400 °C, and (**e**) annealed at 500 °C.

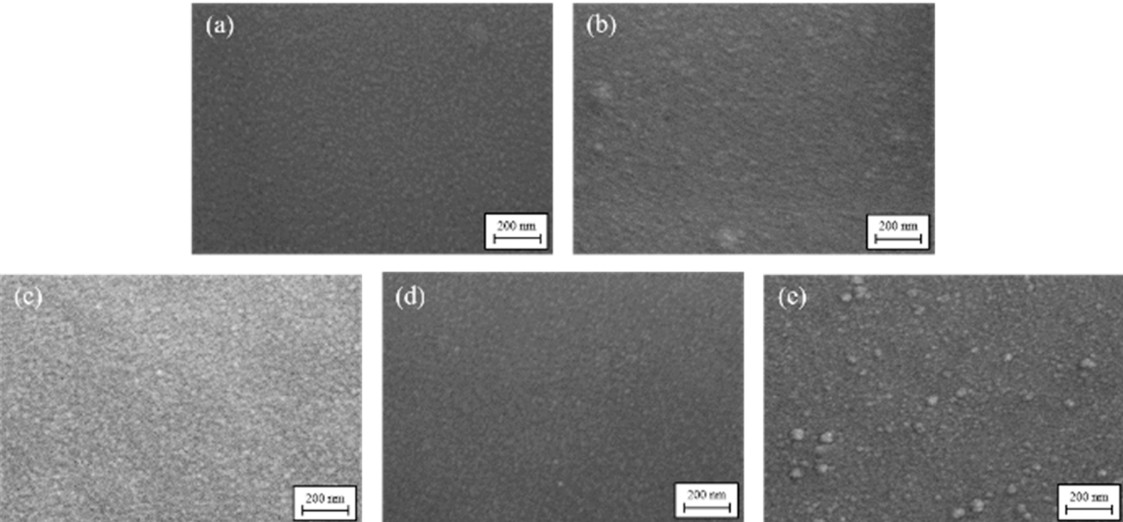

**Figure 6.** SEM images of as-deposited and annealed Ti:SnO$_2$ films: (**a**) as-deposited, (**b**) annealed at 200 °C, (**c**) annealed at 300 °C, (**d**) annealed at 400 °C, and (**e**) annealed at 500 °C.

**Table 3.** Effects of annealing temperature on surface roughness of Ti:SnO$_2$ films.

| Annealing Temperature (°C) | *Ra* (nm) |
|:---:|:---:|
| as-deposited | 3.10 ± 0.02 |
| 200 | 3.23 ± 0.02 |
| 300 | 3.50 ± 0.02 |
| 400 | 3.03 ± 0.02 |
| 500 | 2.96 ± 0.02 |

*3.6. Effects of Annealing Temperature on Crystal Size*

The full width at half maximum (FWHM) values of the peaks in the XRD patterns can be derived from the following Scherrer formula [35]:

$$D = 0.9 \times \lambda/\beta\cos\theta \tag{2}$$

where $D$ is the grain size, $\beta$ is the XRD peak FWHM, $\lambda$ is the wavelength of the incident light, and $\theta$ is the diffraction angle of the incident light. In the XRD process, $\lambda$ and $\theta$ have constant values. Consequently, the grain size, $D$, and FWHM, $\beta$, are inversely related. (Cullity and Stock 2001). Figure 7 shows the FWHM and crystal grain size values of the present Ti:SnO$_2$ films. Note that the as-deposited film has a small crystal structure, and hence the FWHM and crystal size values are also calculated carefully. However, for an annealing temperature of 300 °C, the Ti:SnO$_2$ film has a crystalline structure with a grain size of around 14.89 nm. As the annealing temperature is increased, the crystalline structure becomes more pronounced. Consequently, the grain size decreases, while the FWHM increases. For the maximum annealing temperature of 500 °C, the grain size is equal to approximately 11.56 nm, while the FWHM increases to 0.8.

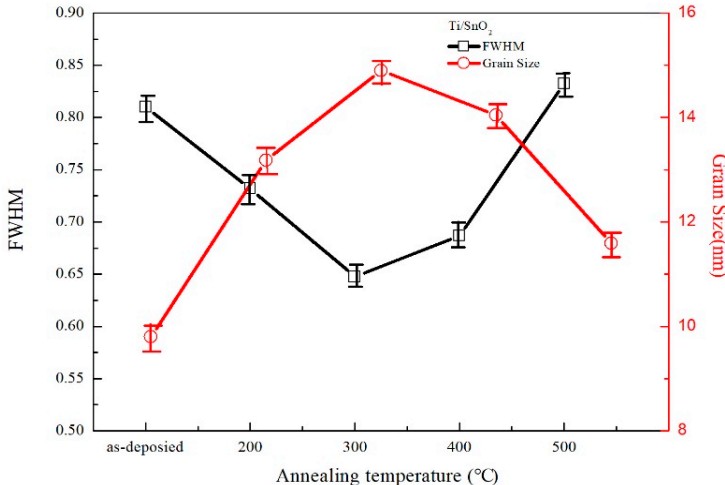

**Figure 7.** The full width at half maximum (FWHM) and grain size of as-deposited and annealed Ti:SnO$_2$ films.

### 3.7. Figure of Merit($\Phi_{TC}$)

The figure of merit ($\Phi_{TC}$) is an important factor used to evaluate the performance of TCO films from the relationship between transmittance and electrical properties. The figure of merit ($\Phi_{TC}$) is defined as [36] as follows:

$$\Phi_{TC} = T^{10}/R_{sh} \tag{3}$$

where $T$ is the average optical transmittance, and $R_{sh}$ is the sheet resistance of the films. Table 4 shows the figure of merit($\Phi_{TC}$) results for the as-deposited and annealed Ti:SnO$_2$ thin films. As shown, the optimal $\Phi_{TC}$ ($3.99 \times 10^{-4}$ $\Omega^{-1}$) is obtained at an annealing temperature of 500 °C. According to the figure, we see that because of an insignificant difference in electrical properties, it has led to a more obvious influence on quality elements from mean optical transmittance, and the optimal mean transmittance is seen at 500 °C. Therefore, optimum quality elements can be achieved at 500 °C.

**Table 4.** The figure of merit of Ti/SnO$_2$ ($\Phi_{TC}$ ($\Omega^{-1}$)).

| Annealing Temperature (°C) | Ti/SnO$_2$ |
|:---:|:---:|
| as-deposited | $1.61 \times 10^{-5}$ |
| 200 | $2.79 \times 10^{-5}$ |
| 300 | $1.99 \times 10^{-4}$ |
| 400 | $2.27 \times 10^{-4}$ |
| 500 | $3.99 \times 10^{-4}$ |

### 3.8. Comparison of Other Methods to the Ti-Doped SnO$_2$ Method

Table 5 shows some literature about the Ti-doped SnO$_2$ thin film under different methods (e.g., sol-gel and ultrasonic spray). We listed the optical and electrical properties in Table 5 from the literature and this study. From Table 5, it can be seen that the RF sputter method has some better optical and electrical properties than sol-gel. However, the cost of the sputtering method is more expansive.

**Table 5.** A comparison of the values of film properties in this study with other methods.

| Method | Transmittance (%) | Resistivity ($\Omega\cdot$cm) | Band Gap(eV) |
|---|---|---|---|
| Rf-Sputter (this study, the doped content of Ti is at 5 at %) | Average is 74.2% (annealing at 500 °C, maximum is 92%) | $5.65 \times 10^{-3}$ (annealing at 300 °C) | 3.21 (annealing at 500 °C) |
| Ultrasonic spray [37] | Maximum 83% (the doped content of Ti is at 4 at %) | $7.64 \times 10^{-3}$ (the doped content of Ti is at 4 at %) | 3.91 (the doped content of Ti is at 4 at %) |
| Sol-gel [38] | none | 62.5 (calcined at 1000 °C) | none |
| Sol-gel [39] | Maximum 74% (the doped content of Ti is at 5 at %) | None | 3.67 (the doped content of Ti is at 5 at %) |

## 4. Conclusions

This study has examined the electrical and optical properties of Ti:SnO$_2$ thin films deposited on glass substrates and then annealed at temperatures ranging from 200–500 °C. The experimental results have shown that the thickness of the Ti:SnO$_2$ films is insensitive to the annealing temperature. However, as the annealing temperature increases, strong peaks in the XRD patterns emerge corresponding to (110), (101), and (211) phases. Hence, it is inferred that the films increase a well-crystalline structure at higher annealing temperatures. The Ti:SnO$_2$ film annealed at the lowest temperature of 300 °C shows both the minimum resistivity of $5.65 \times 10^{-3}$ $\Omega\cdot$cm. The energy gap and optical transmittance both increase with increasing annealing temperature and have values of 3.28 eV and 74.2% at an annealing temperature of 500 °C. The AFM results show that for the samples annealed at temperatures of more than 300 °C, the mean surface roughness reduces with an increase in annealing temperature. The SEM observations suggest that the lower surface roughness is the result of larger grain size. In particular, the grain size decreases from 14.89 nm in the film annealed at 300 °C to 11.56 nm in the film annealed at 500 °C. The Ti:SnO$_2$ film annealed at a temperature of 500 °C shows the highest $\Phi_{TC}$ of $3.99 \times 10^{-4}$ $\Omega^{-1}$. The characterization results have suggested that the optimal performance of this film is due to optical transmittance.

**Author Contributions:** Data curation—formal analysis, C.-F.L. and Y.-S.H.; Writing—Original draft preparation—methodology—investigation T.-H.C.; Review and editing—investigation C.-H.K. All authors have read and agreed to the published version of the manuscript.

**Funding:** This research was funded by the Ministry of Science and Technology, Taipei, Taiwan, under Grant No. 106-2628-E-992 -302 -MY3.

**Acknowledgments:** The authors gratefully acknowledge the financial support provided to this study by the Ministry of Science and Technology.

**Conflicts of Interest:** The authors declare no conflict of interest.

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
