# Peer review of "Optoelectronic Properties of Ti-doped SnO2 Thin Films Processed under Different Annealing Temperatures"

_coatings, doi:10.3390/coatings10040394_

Round 1

Reviewer 1 Report

The paper is well-written. If possible to change the background of figure 4.

Reviewer 2 Report

  1. Section 2. There is no any infromation about the content of Ti. This information must be provided.
  2. What is the reason for such considerable increase in the film thickness at 200 degrees? Such increase cannot be considered as insignificant (Section3.1).
  3. Section 3.3 header: correct of -> on
  4. Lines 128-129: the conclusion "...a crystalline structure becomes more complete (see Fig. 1), and hence the mean transmittance increases" is not supported in any way. This is quite doubtful.
  5. Section 3.4, Table 2. It is not clear how the band gap was calculated.
  6. Results presented in Section 3.5 cause a big concern. What is Ra in Table 3? If this is rms roughness then all values in this table are almost the same and thier differences are definetely in the range of measurement accuracy. At the same time the images in Fig. 4 are noticeably different. All values in Table 3 must be re-checked. The conclusion (lines 153-154) "following annealing at temperatures of 300℃ or more, the surface has a low roughness due to the recrystallize" is again very doubtful.
  7. The heading of Table 4 and the text in this table must be checked and brought in correspondence.
  8. Section 3.7. It should be indicated how the figure of merit is introduced.

Reviewer 3 Report

In this paper, the authors present the effects of the annealing temperature on optoelectronic properties of Ti-doped SnO2 prepared by an RF magnetron sputtering system. However, below are listed some major issues.

  1. The entire text should be carefully checked for language errors.

For instance,

 Page 3, line 103-104, “Consequently, t. However, as the annealing temperature is increased to 300℃ he film has a low resistivity of 5.65x10-3Ω-cm as a result of the high carrier concentration.”  

Page 4, line 105, “Meantime, as the annealing temperature is increases beyond 300℃, SnO and SnO2 combined phase are gradually formed; causing the carrier concentration to decrease and the resistivity to increase”

  1. Missing target information? Sputtering power, flow rate also influences the thin film properties. Is there any reason for 60 W?
  2. In section3.1, only one measurement was measured for each condition. Several measurements (at least 5) need to be measured for each condition and provide average value and error analysis should be included. 
  3. In page 3, lines 87-88, authors state that “Notably, diffraction peaks are not observed in the as-deposited film or the film annealed at 200℃.”, which is a wrong statement. As shown in Figure 1, there is a diffraction peak for as-deposited sample around 20-25 range. Need to be analyzed again.
  4. In Figure 2, y-axis unit is wrong. Error analysis is also missing.
  5. Optical band gap calculation graphs are missing.
  6. In line 142, not mV, should be eV.
  7. Figure 4 scale bar is difficult to see. The black background should be removed and provide a better resolution Figure.
  8. The data value for as-deposited sample is missing in Figure 6.
  9. The equation for the figure of merit is missing.

  10. It is completely unclear how many samples were prepared for each condition, and no error analysis is reported for thickness measurement, surface roughness and the electrical properties (carrier concentration, mobility, and resistivity). Error analysis and numbers of runs need to be provided throughout the paper.

    12. The comparison of values of resulting film properties with previously reported works prepared by different techniques should be added so the reader gets an idea of how much this work advances.

I would not recommend this manuscript to be published at this stage.

Round 2

Reviewer 2 Report

In lines 128-129: the doubtful conclusion "...a crystalline structure becomes more complete (see Fig. 1), and hence the mean transmittance increases" was replaced by the entirely wrong conclusion "The literature [33] reported that light scattering can be caused by surface micro-roughness and means that we can attribute the increased optical transmittance of the annealed Ti:SnO2 films to the flatter surface."

The surface micro-roughness presented in Fig.6 is SMALL SCALE ROUGHNESS that CANNOT cause light losses by scattering. It is better not to provide any interpretation to this part of the text than to provide a wrong one.

Reviewer 3 Report

The authors have addressed the majority of the comments from the reviewers.  I suggest it must go through some revisions below before it can be accepted for publication.

  • The optical band gap (Eg) was calculated from equation 1. It should be with some constant otherwise doesn’t make sense for the unit both sides. Please refer and cite this paper for more information DOI:1007/s10854-016-5223-9
  • Figure 4 unit is not (a.u.). It should be (cm-2eV2).
  • Figure 4 captions should be changed.
  • All parameters in equation 3 of the figure of merit should be defined.
  • Missing superscript and subscript throughout the paper including in the title.
